# Immunotherapeutic Efficacy of IgY Antibodies Targeting the Full-Length Spike Protein in an Animal Model of Middle East Respiratory Syndrome Coronavirus Infection

**DOI:** 10.3390/ph14060511

**Published:** 2021-05-26

**Authors:** Sherif A. El-Kafrawy, Aymn T. Abbas, Sayed S. Sohrab, Ashraf A. Tabll, Ahmed M. Hassan, Naoko Iwata-Yoshikawa, Noriyo Nagata, Esam I. Azhar

**Affiliations:** 1Special Infectious Agents Unit, King Fahd Medical Research Center, King Abdulaziz University, Jeddah 21589, Saudi Arabia; saelkfrawy@kau.edu.sa (S.A.E.-K.); ssohrab@kau.edu.sa (S.S.S.); msahmed@kau.edu.sa (A.M.H.); 2Department of Medical Laboratory Technology, Faculty of Applied Medical Sciences, King Abdulaziz University, Jeddah 21589, Saudi Arabia; 3Biotechnology Research Laboratories, Gastroenterology, Surgery Centre, Mansoura University, Mansoura 35516, Egypt; 4Microbial Biotechnology Department, Genetic Engineering and Biotechnology Division, National Research Centre, Dokki 12622, Egypt; aa.tabll@nrc.sci.eg; 5Department of Immunology, Egypt Center for Research and Regenerative Medicine (ECRRM), Cairo 11517, Egypt; 6Department of Pathology, National Institute of Infectious Diseases, Tokyo 208-0011, Japan; inok@nih.go.jp (N.I.-Y.); nnagata@nih.go.jp (N.N.)

**Keywords:** MERS-CoV, egg yolk antibodies, antiviral, S-protein, in vivo, in vitro

## Abstract

Identified in 2012, the Middle East respiratory syndrome coronavirus (MERS-CoV) causes severe and often fatal acute respiratory illness in humans. No approved prophylactic or therapeutic interventions are currently available. In this study, we developed chicken egg yolk antibodies (IgY Abs) specific to the MERS-CoV spike (S) protein and evaluated their neutralizing efficiency against MERS-CoV infection. S-specific IgY Abs were produced by injecting chickens with the purified recombinant S protein of MERS-CoV at a high titer (4.4 mg/mL per egg yolk) at week 7 post immunization. Western blotting and immune-dot blot assays demonstrated specific binding to the MERS-CoV S protein. In vitro neutralization of the generated IgY Abs against MERS-CoV was evaluated and showed a 50% neutralizing concentration of 51.42 μg/mL. In vivo testing using a human-transgenic mouse model showed a reduction of viral antigen positive cells in treated mice, compared to the adjuvant-only controls. Moreover, the lung cells of the treated mice showed significantly reduced inflammation, compared to the controls. Our results show efficient neutralization of MERS-CoV infection both in vitro and in vivo using S-specific IgY Abs. Clinical trials are needed to evaluate the efficiency of the IgY Abs in camels and humans.

## 1. Introduction

Respiratory infections affect millions of people worldwide and pose risks to many, especially children and the elderly [1]. Middle East respiratory syndrome coronavirus (MERS-CoV) is an emerging zoonotic virus causing severe and often fatal respiratory illness in humans [2]. MERS-CoV was first detected in 2012 [3,4]. Since then, documented infections in humans have steadily increased, with 2566 cases as of December 2020 and an estimated 35% fatality rate [5]. The virus can transmit from camel to camel, and dromedary camels demonstrate high seropositivity to MERS-CoV [6,7,8]. Transmission from camel to human also occurs, and several risk factors, such as direct contact with infected dromedary camels, have been identified [9,10,11,12]. Importantly, MERS-CoV remains endemic in the Middle East. However, it may have pandemic potential, as it has been introduced in other countries via air travel, including an outbreak in South Korea involving more than 100 cases [13]. Presently, there are no approved treatments for MERS-CoV [14] or vaccines to prevent it in humans or camels [15] Therefore, it is important to devise novel antiviral strategies to combat the spread of infection [16].

Thus far, the most promising treatment is the passive administration of anti-MERS-CoV neutralizing antibodies. Several research groups have developed and produced anti-MERS patient-derived or humanized monoclonal neutralizing antibodies in vitro that can protect MERS-CoV-infected mice [2,17,18,19]. These antibodies react with a single epitope on the MERS-CoV spike (S) protein, which is prone to mutations, thus raising the possibility of antibody escape [17,20]. A previous study of passive immunotherapy found that camel serum significantly reduced virus loads and accelerated virus clearance from the lungs of MERS-CoV-infected mice [21]. In another study, equine immunoglobulin-derived F(ab’)2 fragments administered to MERS-CoV-infected mice yielded similar results [22].

Immunoglobulin Y (IgY) is the primary antibody in oviparous animals [23]. It is the only antibody transferred to the egg yolk, from which it can be easily isolated using precipitation techniques [24]. In recent years, IgYs have drawn considerable attention as potential alternatives for passive immunization [25,26,27]. IgYs are safer than antibodies from other species such as IgGs because they do not bind to human Fc receptors or fix mammalian complement components; hence, they do not trigger potentially dangerous immune responses [28]. Chickens can produce eggs with IgY antibodies on a large scale using non-invasive and humane methods, which may offer new, economically feasible, and efficient immunotherapy options [29,30,31,32,33]. Furthermore, IgY has greater binding avidity to target antigens than mammalian IgG [34], and it can be produced against conserved mammalian proteins more easily and more successfully than IgG can be produced in other mammals due to the evolutionary distance between mammals and birds [24]. IgYs also induce an efficient immune response in low quantities [31].

Specific IgY antibodies have proven highly effective for the prevention and treatment of respiratory viral and bacterial diseases such as influenza A [35,36,37,38], influenza B [39], SARS coronavirus [40], bovine respiratory syncytial virus [41], and *Mycobacterium tuberculosis* (TB) infection [42]. IgY technology has been successfully applied in clinical trials against *Pseudomonas aeruginosa* lung infection [43]. In 2008, the European Medicines Agency granted an orphan drug designation to IgY for the treatment of cystic fibrosis [44]. Another recent study demonstrated that IgY antibodies transiently decrease *P. aeruginosa* colonization in the airways of mechanically ventilated piglets [45]. Moreover, specific IgY antibodies could protect mice against pneumonia caused by *Acinetobacter baumannii* [44]. IgY antibodies can neutralize viral infectivity by several mechanisms, either by blocking the attachment of the virus to host tissue; by preventing the membrane fusion or promoting the detachment of bound virus; by interfering with free virions; or by causing aggregation of virus particles resulting in virus immobilization [46].

Despite the wide use of chicken for the production of IgY for research, the immunization of other birds for IgY production has also been used with similar immunization protocols. Goose IgY antibodies were generated against dengue, West Nile virus, and Zika Virus and showed antiviral activity in mouse models for these infections [47]. Other studies showed that goose and ducks IgY produced against Andes virus (which is the causative agent of the Hantavirus pulmonary syndrome) showed a prophylactic effect against the virus in infected mice and hamster [48,49]. Ostriches immunized with the swine influenza virus were utilized to generate IgY antibodies against pandemic influenza virus IgY antibodies can be produced in large quantities [38]. Quails were also used to generate anti-*H. pylori* IgY antibodies [50]. The wide use of chicken for the production of IgY antibodies might be due to the widespread and economic production of chicken eggs in large farms.

The MERS-CoV S protein engages with the viral cellular receptor dipeptidyl peptidase 4 (DPP4) to mediate viral attachment to host cells and subsequent fusion of the virus with the cell membrane [18,51,52,53]. The S protein plays a key role in counteracting coronavirus infection, as shown in studies on human-neutralizing antibodies from rare memory B cells in individuals infected with SARS-CoV [54] or MERS-CoV [17]. In such studies, antibodies targeting the S protein of SARS-CoV effectively inhibited virus entry into host cells. More recently, it has been found that SARS-CoV S elicits polyclonal and vigorously neutralized SARS-CoV-2 S-mediated entry into cells, thus encouraging the use of this molecular target for vaccination and immunotherapies [55]. In this study, we continue our previous investigation on the efficacy of IgY in neutralizing MERS-CoV [27] by reporting the first in vitro and in vivo investigations of anti-MERS-CoV S1 IgY antibodies in neutralizing the virus. Together, these two studies are the first to investigate the potential of MERS-CoV-specific IgY to treat MERS-CoV infection in camels and humans.

## 2. Results

### 2.1. Isolation and Purification of IgY

SDS–PAGE revealed that the IgY preparation dissociated into a major and minor protein band with molecular weights of ~68 kDa (heavy chain) and ~27 kDa (light chain), respectively, and a purity of 90% (Figure 1). The total IgY contained in a milliliter of egg yolk was estimated to be 4.4 mg, or about 60 mg of total IgY from a single egg yolk (~15 mL).

### 2.2. Dynamics of Anti-S IgY Antibodies in the Sera of Chickens and Egg Yolks

Steady increases in serum levels of MERS-CoV S-specific IgY titers were observed in chicken sera after the first immunization. Levels peaked in week 7 and remained high until week 12. Sera of chickens who received the adjuvant only showed no reactivity to the MERS-CoV S antigen. Anti-MERS-CoV S antibody titers were not detected in the eggs until week 3 after immunization, then they increased until reaching a peak at week 7, and then plateaued until week 12 (Figure 2).

### 2.3. Immunoreactivity of Anti-S IgY of the MERS-COV

The specificities of anti-MERS-CoV S IgY antibodies were tested using Western blotting analysis. IgY induced by the S protein recognized the recombinant S protein at approximately 142 kDa (Figure 3).

### 2.4. Dot Blotting

The specificities of anti-S IgY antibodies were confirmed by dot blotting analysis. Purified IgY antibodies showed reactivity with the S protein, S1, and receptor-binding domain. They were not reactive to the nucleocapsid protein of MERS-CoV, as shown in Figure 4.

### 2.5. Anti-S IgY Neutralizes MERS-CoV

Anti-S IgY can potently neutralize live MERS-CoV in permissive Vero cells, with 100% neutralization at IC_100_ concentrations less than 12.5 µg/mL. Nonspecific IgY Abs from adjuvant-only controls did not exhibit antiviral activity against MERS-CoV up to 1000 µg/mL (Figure 5). These results suggest that anti-S MERS-CoV IgY antibodies exhibited a potent ability to neutralize MERS-CoV infection. IC_100_ was determined as the reciprocal of the highest dilution at which no CPE was observed in the cells.

### 2.6. RT-qPCR-Based Neutralization Activity

The in vitro neutralization effect of the IgY Abs was examined by mixing different dilutions of the IgY Abs with MERS-CoV incubating for 1 h at 37 °C and then applying to the cells (as described in Section 4.8). This approach showed a high neutralization effect on the virus at a 50% neutralizing concentration (NC_50_) of 51.42 μg/mL. The neutralization effect of the IgY Abs was assessed using real time RT-PCR of the cell cultures treated with different dilutions of the anti-MERS-CoV S IgY Abs, relative to the virus control cells (cells infected with the virus and untreated), which showed concentration-dependent inhibition of the virus (Figure 6). The log IgY concentration was plotted against the percentage of inhibition of each concentration and the NC_50_ was calculated following a nonlinear variable slope equation according to the equation: Y = 100/(1 + 10^((LogIC50-X) × HillSlope))).

### 2.7. IgY Confers In Vivo Protection in Virus-Challenged Mice

MERS-CoV viral titers showed a marked reduction in the quantitative pathological score of the lungs in the anti-S IgY group compared to the controls (Figure 7A) but with no statistically significant difference. The body weights of hDPP4-Tg mice were not significantly different between the MERS-CoV S IgY group and the adjuvant-only group after intranasal inoculation with 10^6^ tissue culture infectious dose 50%(TCID_50_) of MERS-CoV (Figure 7B). Histopathological investigations revealed that Tg mice developed progressive pulmonary inflammation due to acute MERS-CoV infection on day 8 post infection. Inflammatory reactions, including partial and mild cellular infiltration with mononuclear cells and macrophages in response to viral infections, were observed in the alveolar areas of the lung tissues (Figure 7C). Among virus-infected Tg mice, intraperitoneal injection of anti-S IgY antibodies led to significantly weaker inflammatory reactions (*p* = 0.041), compared to the adjuvant-only control group (Figure 7C,D). Immunohistochemistry using an anti-MERS-CoV nucleocapsid polyclonal antibody in lung tissues showed fewer viral-antigen-positive cells in the lungs of the group treated with anti-S IgY (Figure 7E,F), compared with the adjuvant-only controls.

## 3. Discussion

MERS-CoV poses a continuing threat to human health, especially due to its high fatality rate of about 35%. Prevention and treatment strategies to control MERS-CoV infection are urgently needed. Although vaccines remain one of the most important approaches against viral infections, they generally take a long time to develop, and they do not provide immediate prophylactic protection or treat ongoing infections [56]. Passive immunotherapy is a highly successful treatment for some severe and even life-threatening human diseases [57]. Treatments using IgY from chicken eggs has received considerable attention in recent years. Previous studies showed that a single egg can yield up to 100 mg of total IgY, and one hen can produce 250 eggs per year, thus generating large quantities of protective IgY at a comparatively low cost [37].

In this study, hens injected with MERS-CoV S subunit protein were shown to be highly immunogenic, demonstrating a high titer and long-lasting humoral immune response for at least 2 months without the need for boosters. Among treated hens, a high titer of specific MERS-CoV S IgY Abs was observed in the sera at 2 weeks post injection and in the eggs at 4 weeks post injection and remained at this high titer for 12 weeks. Other studies have shown that hens maintain a high antibody titer against a variety of antigens used for immunization for at least 3 to 4 months [37]. A large quantity of high-specificity IgY thus could be produced in a few months using this IgY technology. The results in this study and our previous study [27] indicate the potential for a rapid response to MERS-CoV and other emerging infections [39].

In the Western blot assay, the anti-S MERS-CoV IgY antibody exhibited immunoactivity to viral S recombinant protein, which is reported to promote binding of MERS-CoV to host-cell surface molecules during the attachment phase [58]. Anti-S MERS-CoV IgY Abs also exhibited binding to S1 and receptor-binding domain recombinant proteins. However, a dot-blot immunoassay revealed no reactivity to the recombinant nucleocapsid protein, confirming that anti-S IgY Abs are antigen-specific. This observation aligns with other reported observations that the IgY Abs response to highly conserved mammalian proteins is robust and demonstrates high affinity, meaning it could target a broad spectrum of epitopes on protein immunogens [59]. Moreover, chicken IgY Abs reportedly exhibit higher avidity (10^9^ L/mol) after the first immunization than sheep, which must receive four boosters to reach similar avidity values [60].

The neutralizing activities of the anti-S IgY Abs were assessed in vitro. Vero cells showed dramatic inhibition of MERS-CoV-induced CPE. Quantitative PCR provides a robust, sensitive, and wide dynamic range when used to evaluate antiviral activity [61]. In the present study, qRT-PCR showed a decreased viral load in cells treated with anti-S IgY Abs, compared with virus control cells with no IgY antibodies (50% neutralizing concentration of 51.42 μg/mL). This NC_50_ is comparable to our previous study [27] evaluating the neutralizing effect of anti-S1 IgY Abs against MERS-CoV, which showed a 50% neutralizing concentration of 60 μg/mL in vitro. IgY antibodies are reportedly highly effective in neutralizing other bacterial and viral infections of the respiratory system, with no reported side effects [26,62].

Histopathologic examinations and immunostaining (e.g., immunohistochemistry) of lung tissues are essential to better understand disease pathogenesis and evaluate novel treatments of viral infections (Menachery, Yount et al. 2015, Meyerholz, Lambertz et al. 2016, Cockrell, Johnson et al. 2018, Hua, Vijay et al. 2018). In the case of coronavirus infection, lung histopathology can be a useful tool to define affected cells, illuminate structural causes of clinical signs, and clarify potential therapies (Meyerholz and Beck 2020). In comparison with our previous study [27], we found that MERS-CoV-related inflammation in lung cells decreased significantly in mice treated with MERS-CoV S IgY Abs, compared with the controls. This decrease might be a reflection of the reduced viral antigen positive cells in the lung. Histological reduction of lung tissue inflammation is associated with enhanced viral clearance and rapid recovery of the lung tissue following the transfer of cloned Tc (T cytotoxic cells) [63]. Decreased lung pathology also is associated with IgY antibodies in influenza-infected mice [36,37,39,64]. Our in vivo investigation also showed a marked reduction in viral-antigen-positive lung cells in mice treated with MERS-CoV S IgY Abs, compared with adjuvant-only controls, although this difference was statistically non-significant. Compared to our previous study [27], where we observed a significant reduction in viral-antigen-positive lung cells using MERS-CoV S1 IgY Abs, this study showed a marked but non-significant reduction in the number of antigen positive lung cells in mice treated with anti-S IgY, compared with adjuvant-only controls. As with our previous study, there were no significance change in the body weight of the treated animals compared with controls as well as no significant change in the viral titers.

To date, several anti-MERS-CoV antibodies have been developed, each with advantages and disadvantages. To develop monoclonal antibodies, mouse-derived monoclonal antibodies must be humanized before human use [18]. Human-neutralizing antibodies derived from a convalescent MERS patient can be produced in large quantities from Chinese hamster ovary cells [17]. However, the single-clone antibody raises concerns about viral-escape mutants when applied to humans. In a mouse model of infected lungs, administration of transchromosomic bovine human immunoglobulins [65] or dromedary immune serum [66] leads to rapid viral clearance. These animals are not readily available, though, and several monoclonal antibodies might be needed to induce effective viral clearance. The use of IgY helps in reducing the risk of escape mutants as they target multiple epitopes making it harder for the virus to escape all the targeted positions.

Chickens offer several advantages over conventional mammalian species in producing pathogen-specific antibodies. These include high rates of egg production, high IgY content per egg yolk [67,68,69], and humane and non-invasive methods of collecting IgY Abs from eggs [70,71,72]. Clinical and laboratory data demonstrate that IgYs may offer a safe and effective tool for controlling and treating viral diseases. They may be used as a substitute for or essential complement to antimicrobials and vaccines [73,74,75,76]. In our study, the production of MERS-CoV-specific IgY Abs took 2–3 months, plus 2 additional months for the in vitro and in vivo investigations. This quick timetable makes this approach suitable for responding quickly to emerging and re-emerging pathogens.

## 4. Material and Methods

### 4.1. Immunization of Laying Hens

Eight Lohmann laying hens (25 weeks old) provided by a local broiler farm (Algharbia Breeding Company, Saudi Arabia) were used for egg production. Animals were placed in broiler chicken cages (two animals per cage) in a 12-h light-dark cycle at room temperature (24 ± 3 °C). Water and commercial laying hen food were offered ad libitum. The immunization group (*n* = 4) was injected with 200 µg of recombinant MERS-CoV S protein obtained from Sino Biological, Inc. (Beijing, China). Injections were administered in the left or right side of the pectoral muscle on days 0, 14, 28, and 49. Before each immunization, the recombinant protein was emulsified in a 1:1 ratio with Freund’s Complete Adjuvant (Sigma, St. Louis, MO, USA) for the first immunization, and Freund’s Incomplete Adjuvant (Sigma, St. Louis, MO, USA) was similarly used for subsequent booster immunizations. The suspension was mixed by pipetting up and down in a 19-gauge needle attached to a 5-mL syringe until stable. The control group (*n* = 4) was injected with phosphate-buffered saline (PBS) plus the corresponding adjuvant. Blood samples were taken before each injection and on the day before slaughter. Eggs were collected daily 1 week before the initial immunization and continued for 12 weeks after immunization. Eggs were stored at 4 °C to isolate IgY from the yolk. The Biomedical Ethics Research Committee of the Faculty of Medicine at King Abdulaziz University reviewed and approved the experimental protocol (permit no.: 120-18).

### 4.2. Isolation and Purification of Yolk IgY

Egg yolks from the harvested eggs of immunized and non-immunized hens were pooled and separated from egg whites using egg separators and then washed with deionized water. IgY purification was performed using a Pierce Chicken IgY Purification Kit (Thermo Fisher Scientific, Waltham, MA, USA). IgY concentration was determined via spectrophotometry measuring absorbance at 280 nm (A280) according to the manufacturer’s instructions.

### 4.3. Sodium Dodecyl Sulfate–Polyacrylamide Gel Electrophoresis

Sodium dodecyl sulfate–polyacrylamide gel electrophoresis (SDS–PAGE) was performed to determine the purity and molecular weight of IgY using 12% PAGE with a Mini-PROTEAN^®^ 3 cell (Bio-Rad Laboratories, Hercules, CA, USA). The analysis was conducted under reducing conditions: the sample was mixed with 2× sample buffer boiled for 10 min at 100 °C, then 25 μL of purified IgY was loaded into each well. Prestained Blue Protein Marker (MOLEQULE-ON, Auckland, New Zealand) was used as a molecular weight marker. Electrophoresis was performed at room temperature in running buffer (Tris-glycine buffer) at 200 volts for 40 min. Protein bands were visualized using Coomassie Brilliant Blue stain (Abcam, Cambridge, UK) and analyzed using Gene Tools image analysis software (Syngene, Cambridge, UK).

### 4.4. Reactivity of Anti-S IgY Antibodies by ELISA

The antibody reactivity of anti-S IgY was determined by ELISA. Briefly, microtiter plates were coated with purified MERS-CoV-S antigen (Sino Biological, Inc., Beijing, China) at 500 ng/mL in PBS (0.01 M, pH 7.4) at 100 μL/well and then stored at 4 °C overnight. After washing the plates once with PBS and twice with Tween-20, they were blocked with 250 μL of blocking buffer (5% skim milk in PBS-Tween) at room temperature for 1 h. The wells were washed three times with wash buffer. IgY antibody titers were determined by serially diluting the serum and purified IgY from immunized and non-immunized hens, starting with a 1:50 ratio in blocking buffer. The plates then were incubated at 37 °C for 1 h and washed three times with PBS-Tween. A 1:10,000 dilution of horseradish peroxidase (HRP)-conjugated rabbit anti-chicken IgY (Abcam, Cambridge, UK) was added to each well (100 μL/well) and incubated for 1 h at 37 °C. After washing the plates, the color reaction was developed by adding TMB (100 μL/well) substrate solution (Promega, Madison, WI, USA) and incubating for 30 min. This reaction was stopped by adding 2M H_2_SO_4_ (100 μL/well).

The optical density (OD) of each well was read at 450 nm using a microtiter plate reader (ELX800 Biokit). PBS was used as a blank control, and purified IgY derived from non-immunized hens was used as a negative control. The titer of anti-S IgY was defined as the maximum dilution of the sample that resulted in an OD value 2.1 times higher than that of the negative control.

### 4.5. Western Blotting Assay

Western blotting was performed to check the specificity of the anti-MERS-CoV S IgY antibody using a previously described method with some modifications [77]. Five µL containing 500 ng of recombinant S protein was mixed with 20 µL of electrophoresis sample buffer and then subjected to SDS–PAGE in a 14% slab of polyacrylamide gel separated by a 4% stacking gel at 200 V for 40 min at room temperature. The gel and blotting papers were equilibrated in transfer buffer for 10 min, after which the S protein was electrically transferred onto a polyvinylidene fluoride (PVDF) membrane activated by methanol (Thermo Fisher, Waltham, MA, USA) at 30 V overnight. The PVDF membrane was cut into 0.5-cm strips, which were blocked with Tris-buffered saline containing 0.1% Tween 20 (TBS-T) and 5% non-fat dry milk for 1 h at room temperature. The strips were washed three times for 10 min each. The membrane was then incubated in a 1:50 dilution of anti-MERS-CoV S IgY antibodies. After incubation, the strips were washed three times with TBS-T for 10 min each and incubated with HRP-conjugated rabbit anti-chicken IgY Heavy and Light (Abcam, Cambridge, UK) at a 1:10,000 dilution in blocking buffer for 1 h at room temperature. The strips again were washed three times for 10 min, after which they were incubated with HRP colorimetric substrate (Immun-Blot Opti-4CN colorimetric Kit, Bio-Rad) for 15 min at room temperature. This reaction was stopped by rinsing with distilled water. The strips were photographed after development. The same Western blotting procedure was performed to identify the presence of the S IgY antibodies. This was done by subjecting the anti-MERS-CoV S IgY antibodies to SDS-PAGE, transferring onto PVDF membrane, followed by addition of HRP-conjugated rabbit anti-chicken IgY Heavy and development by adding HRP colorimetric substrate.

### 4.6. Dot-Blotting

A dot-blot assay was performed to determine the specificity of the purified anti-S IgY antibodies. PVDF membranes were activated by soaking in methanol for 15 s and washing with distilled water. Then, three different concentrations (500, 100 and 50 ng) of the recombinant antigens S, S1, nucleocapsid, and PBD were dot-blotted individually onto a PVDF membrane. The membrane was incubated in 20 mL of blocking buffer for 1 h at room temperature. After washing three times with TBS-T, the PVDF membrane was immersed in primary anti-MERS-CoV S IgY antibodies (1:200 dilution) in blocking buffer with gentle agitation for 1 h at room temperature. The membrane was incubated with rabbit anti-chicken IgY HRP-conjugate as a secondary antibody (1:10000 dilution) in blocking buffer with gentle agitation for 1 h at room temperature. After washing as previously described, the membrane was placed on an HRP colorimetric substrate (Immun-Blot Opti-4CN Colorimetric Kit, Cat. No. 1708235) (Bio-Rad Laboratories, Hercules, CA, USA) for up to 30 min at room temperature. The reaction was stopped using distilled water.

### 4.7. Microneutralization Assay

Live virus experiments were performed in a biosafety level 3 laboratory in the infectious agent unit of King Fahd Medical Research Center at King Abdulaziz University in Jeddah. A neutralizing assay was performed, as previously described [27,78]. Briefly, MERS-CoV isolate at an MOI of 0.01 (500 μL) in the presence or absence of IgY antibodies was added to an equal volume of serial dilutions of the IgY antibodies for 1 h. The mixture was then inoculated in triplicate onto Vero E6 cells (10,000 cells/well) on 96-well plates and in viral inoculation medium (Dulbecco’s Modified Eagle Medium with 2% fetal bovine serum, 1% penicillin/streptomycin, and 10 mmol/L HEPES at pH 7.2). Cells were incubated in a humidified incubator with 5% CO_2_ at 37 °C for 2–3 days or until reaching an 80–90% cytopathic effect (CPE) in positive virus control wells (virus with no added IgY Abs). The IC_100_ neutralization of the antibody was determined as the reciprocal of the highest dilution at which no CPE was observed.

### 4.8. Neutralization Using Real-Time qRT-PCR

The MERS-CoV isolates at an MOI of 0.01 (500 μL) were added to an equal volume of varying dilutions of the IgY antibodies (440, 220, 110, 55, 44, 22, and 11 μg/mL). The mixture was then inoculated onto Vero cells (10,000 cells/well in triplicate) on 96-well plates and in the previously described viral inoculation medium. Cells were incubated in a humidified incubator with 5% CO2 at 37 °C for 2–3 days or until reaching 80–90% CPE in positive virus control wells (virus with no added IgY Abs). Upon reaching 80–90% CPE in control wells, 200 μL of culture supernatants were collected, cleared by centrifugation (500× *g*, 5 min, 4 °C), and stored at −70 °C. In each experiment, a negative control with no added virus or IgY was included.

Real-time RT-qPCR was performed using primers and probes targeting the MERS-CoV N gene, as previously described [78], to assess the neutralization effect of the IgY antibodies. A neutralizing concentration of 50% was used to express IgY Ab neutralization activity and to define the concentration of IgY Ab needed to reduce the viral RNA copies by 50%, relative to the positive virus control.

### 4.9. Effect of Anti-S IgY Antibodies in Transgenic Mice Infected with MERS-CoV

A mouse model of MERS-CoV was used in this study, as previously described [27,79] Yoshikawa et al., JV, 2019. Briefly, transgenic (Tg) mice on a C57BL/6NCr (SLC, Inc., Hamamatsu, Japan) background were developed to express human CD26/dipeptidyl peptidase 4 (hDPP4), a functional receptor for MERS-CoV under the control of an endogenous hDPP4 promoter. The hDPP4-Tg mice (*n* = 10) were intranasally infected with MERS-CoV using the HCoV-EMC 2012 strain (10^6^ TCID_50_) provided by Dr. Bart Haagmans and Dr. Ron Fochier (Erasmus Medical Center, Rotterdam, the Netherlands). Mice also received a peritoneal injection of either 500 μg of anti-S IgY antibodies or 500 μg of IgY isotype control at 6 h and 1 day post infection. The animal experiment was conducted simultaneously with that described in a previous report of the efficacy of a MERS-CoV anti-S1 IgY antibody [27]. Thus, data from the control mice are used in both studies. Mouse weight was measured at 8 days post infection.

Animals were sacrificed at 1-, 3-, or 5-days post infection (*n* = 4), and lung tissues were collected for virological detection. After 8 days of observation, the remaining 6 mice were sacrificed for histopathological evaluations. All work with MERS-CoV and passive immunization of mice was conducted at the National Institute of Infectious Diseases in Tokyo, Japan. Stocks of MERS-CoV were propagated and titrated on Vero E6 cells and cryopreserved at −80 °C. Viral infectivity titers were expressed as the TCID_50_/_mL_ on Vero E6 cells and calculated according to the Behrens-Kärber method. Work with infectious MERS-CoV was performed under biosafety level 3 conditions.

### 4.10. Histopathology and Immunohistochemistry

After anesthetizing and perfusion with 2 mL of 10% phosphate-buffered formalin, the mouse lungs were harvested and fixed in paraffin, sectioned, and subjected to hematoxylin and eosin staining. The tissue sections then were autoclaved at 121 °C for 10 min in a retrieval solution at pH 6.0 (Nichirei Biosciences Inc., Tokyo, Japan) for antigen retrieval in preparation for immunohistochemistry. MERS-CoV antigens were detected using a polymer-based detection system (Nichirei-Histofine Simple Stain Mouse MAX PO(R), Nichirei, Tokyo, Japan) with a rabbit anti-MERS-CoV nucleocapsid antibody (40068-RP01, Sino Biological Inc., Beijing, China). Peroxidase activity was detected using 3,3′-diaminobenzidine (Sigma-Aldrich, St. Louis, MO, USA), and hematoxylin was used for counterstaining.

### 4.11. Quantitative Analysis of Inflammation and Viral Antigen Positivity of Cells

Inflammation was assessed using hematoxylin and eosin staining on the paraffin-embedded sections (3 mm thickness) from the Tg mice at 8 days post infection. Light microscopic images were obtained using a DP71 digital camera under low-power magnification and cellSens software (Olympus Corporation, Tokyo, Japan). Inflammation was evaluated by measuring three lobes with an average section area of 3.645 ± 0.726 mm^2^. The inflammation areas were traced using the contour measurement program Neurolucida (version 12, MBF Bioscience, Williston, VT, USA) and analyzed using Neurolucida Explorer (MBF Bioscience Williston, VT, USA). Viral antigen was detected via immunohistochemistry on a continuous paraffin-embedded section. Cells positive for viral antigen were counted in images under high-power magnification (observation area: 0.147 mm^2^). Data for the control mice came from a previous study assessing the efficacy of anti-S1 MERS-CoV IgY antibodies [27] as the two experiments were performed simultaneously.

### 4.12. Statistical Analysis

Data are expressed as means with standard errors. Statistical analyses were performed using Graph Pad Prism 9 software (GraphPad Software Inc., La Jolla, CA, USA). Intergroup comparisons (virus titers in the lungs and body weight curves) were performed using two-way analyses of variance, followed by Bonferroni’s multiple comparisons test. Comparisons between two groups (the quantitative analysis of inflammation and viral antigen positivity in cells) were performed using the Mann–Whitney test. A *p*-value of <0.05 was considered statistically significant.

### 4.13. Ethics Statement

The Biomedical Ethics Research Committee of the Faculty of Medicine at King Abdulaziz University reviewed and approved the experimental protocol for the immunization and handling of the chickens (permit no.: 120-18). The Committee for Experiments Using Recombinant DNA and Pathogens at the National Institute of Infectious Diseases in Tokyo, Japan, approved the experiments using recombinant DNA and pathogens. Animal studies strictly followed the Guidelines for Proper Conduct of Animal Experiments of the Science Council of Japan and complied with animal husbandry and welfare regulations. All animals were housed in a facility certified by the Japan Health Sciences Foundation. Animal experiments also were approved by the Committee on Experimental Animals at the National Institute of Infectious Diseases in Japan, and all experimental animals were handled in accordance with biosafety level 3 animal facilities according to the committee guidelines.

## 5. Conclusions

The results presented in this study provide evidence for the specific and efficient neutralization of MERS-CoV using anti-S IgY antibodies in vitro and in an animal model of MERS-CoV. Together with our previous study, the two studies provide the first evidence for the potential use of MERS-CoV-specific IgY antibodies as a therapeutic vaccine against MERS-CoV. Further studies are needed to investigate the combined effect of both anti-S and anti-S1 IgY Abs in neutralizing MERS-CoV through intraperitoneal and intranasal routes of administration. Clinical trials are needed to evaluate the efficacy of this therapy in camels and humans. The IgY antibodies might prove useful for treating MERS-CoV in high-risk populations, such as those with immature or weakened immunity, or in high-exposure groups, such as healthcare workers, camel handlers, and slaughterhouse workers. Furthermore, the IgY antibodies can be used to treat MERS-CoV in camels, which can transmit the virus to humans. The data generated in this study provide a platform for future studies to generate specific and efficient IgY antibodies against other coronaviruses.

## Figures and Tables

**Figure 1 pharmaceuticals-14-00511-f001:**
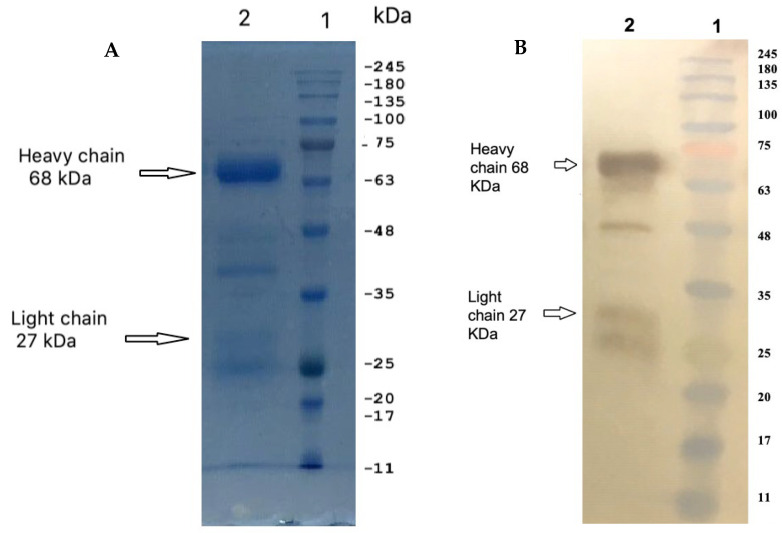
(**A**) SDS-PAGE profile of anti-MERS-CoV S IgY antibodies. The two IgY chains appeared using 10% resolving SDS-PAGE gel. The molecular weight of the heavy chain is 68 kDa, and the molecular weight of the light chain is 27 kDa. (**B**) Western blot identification of IgY using HRP-conjugated rabbit anti-chicken IgY heavy and light. Remaining bands might represent other antibodies or protein fragments of unknown origin.

**Figure 2 pharmaceuticals-14-00511-f002:**
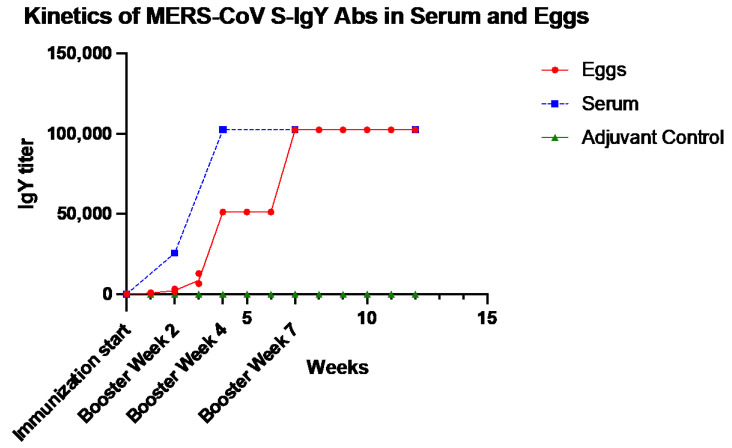
Kinetics of chicken serum and egg yolk antibody response to anti-MERS-CoV S IgY after infection with MERS-CoV S recombinant protein, compared with adjuvant-only controls. Each week is represented by a pool of egg yolks of individual chickens from each group (S-immunized and adjuvant-only).

**Figure 3 pharmaceuticals-14-00511-f003:**
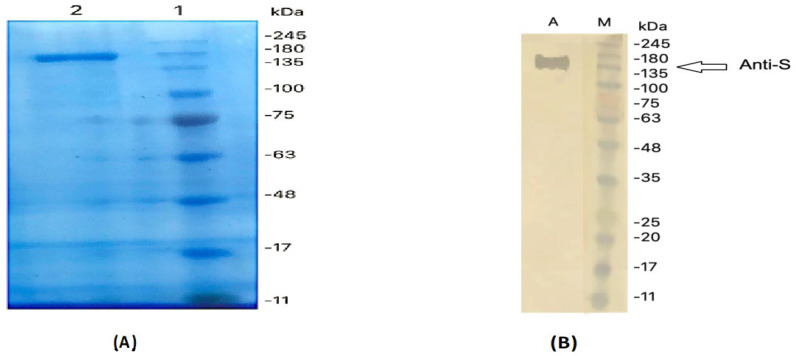
Western blot analysis of anti-MERS-CoV S IgY antibodies. (**A**) The S protein of MERS-CoV subjected to SDS-PAGE under reducing conditions. (**B**) Western blot analysis of the anti-S IgY antibody response.

**Figure 4 pharmaceuticals-14-00511-f004:**
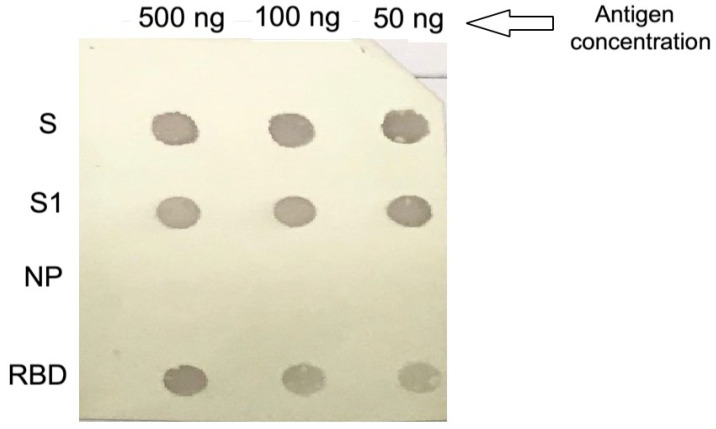
Dot blotting analysis. Purified anti-S IgY antibodies showed reactivity with different concentrations of the S, S1, and receptor-binding domain proteins but had no reactivity with the nucleocapsid protein of MERS-CoV.

**Figure 5 pharmaceuticals-14-00511-f005:**
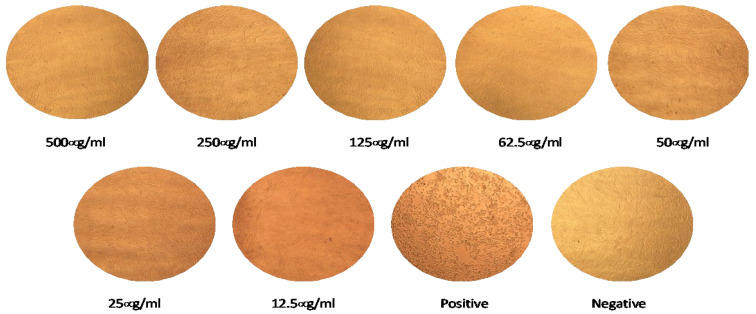
Cytopathic effect of different concentrations of anti-S IgY antibodies against MERS-CoV in Vero-E6 cells. The IC_100_ neutralization of the antibody was determined as the reciprocal of the highest dilution at which no cytopathic effect was observed.

**Figure 6 pharmaceuticals-14-00511-f006:**
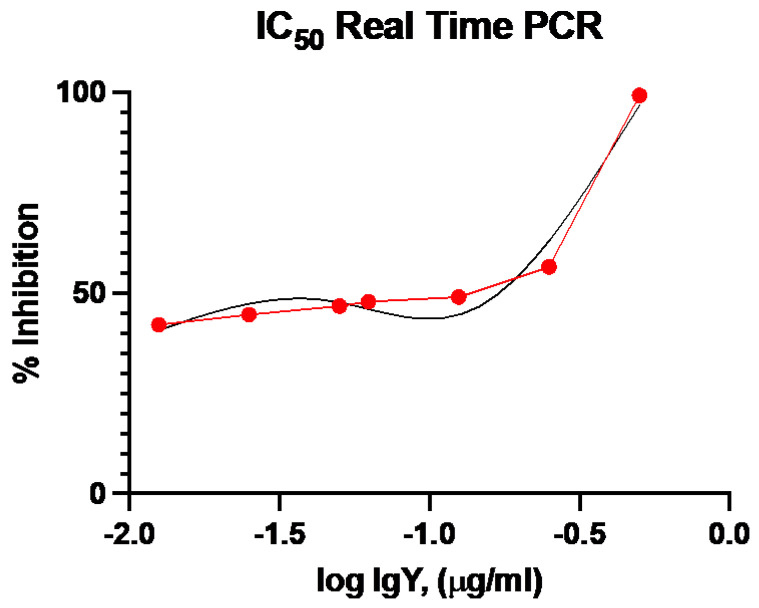
Virus neutralization titer of PCR-based virus neutralization test. IgY concentrations are represented in log_10_ (μg/mL). Red line connects the actual points while the black line represents the curve fitting line.

**Figure 7 pharmaceuticals-14-00511-f007:**
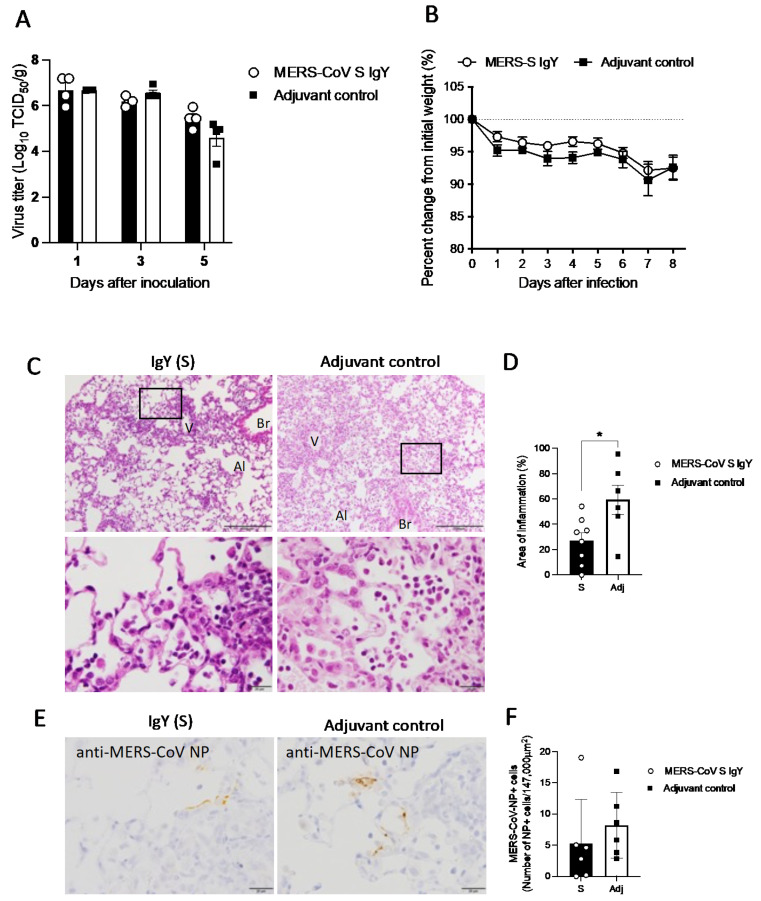
(**A**) Viral titers in lung homogenates of MERS-CoV infected mice at 1-, 3-, and 5-days after inoculation (*n* = 4 per group). Mice were treated with anti-S IgY antibodies or adjuvant only. The detection limit was 10^1.5 TCID_50_/g of tissue. (**B**) Body weight changes between mice with anti-S IgY antibodies and the adjuvant-only controls after MERS-CoV infection. (**C**–**F**) Histopathology of the lungs from human dipeptidyl peptidase 4-transgenic mice on day 8 after infection with MERS-CoV. (**C**) Representative histopathological findings of mice with highest cellular infiltration in the alveoli, identified using hematoxylin and eosin staining. Massive mononuclear cell infiltrations, including macrophages and lymphocytes with regenerated type II pneumocytes, were observed in the control group (right column) but slightly less in the group treated with anti-S IgY (left column). Scale bars: 200 μm (upper row) and 20 μm (lower row). Al, alveoli; Br, bronchi; V, vessel. (**D**) Quantification of inflammation areas. Pulmonary lesion areas were determined based on the mean percentage of affected areas in each section of the collected lobes from six animals. Circles indicate averages from three observation lobes in each mouse (*p* = 0.041 by Mann-Whitney test). (**E**) Detection of viral antigen in the lung tissues of mice, determined by immunohistochemistry. Antigen-positive cells were observed less frequently in the lungs of the group treated with anti-S-IgY, compared to the adjuvant-only controls. Scale bars: 20 μm. (**F**) Numbers of viral-antigen-positive cells in the alveoli from six mice. Circles indicate averages of five observation fields in each mouse (*p* = 0.258 by Mann-Whitney test). The asterisk indicates statistical significance.

## Data Availability

All data for this article are available in manuscript.

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
