# Peer review of "Immunotherapeutic Efficacy of IgY Antibodies Targeting the Full-Length Spike Protein in an Animal Model of Middle East Respiratory Syndrome Coronavirus Infection"

_pharmaceuticals, 2021, doi:10.3390/ph14060511_

Round 1
Reviewer 1 Report
The authors present both very interesting and timely findings of the ability to rapidly raise MERS-CoV (MERS) S protein-specific IgY, present in large amounts. This MERS IgY recognized the S protein, and was neutralizing for the MERS-CoV virus both in vitro and in vivo. This is proposed to provide a passive immunotherapy both for MERS infections in humans and camels, as well as additional support for the potential of IgY immunotherapeutics.
A major deficiency in the manuscript is the absence of discussion or hypothesis of the mechanism of protection by IgY. It has been published repeated that IgY is unique from mammalian immunoglobulins in that IgY does not bind to any mammalian FcRs, mammalian complement, rheumatoid factor, etc. While it can be argued that this provides an excellent niche for IgY because of the absence of inflammation associated with these various agents, this then makes the route of IgY neutralization unclear. This should be discussed in the Introduction (and/or possible the Discussion).
A second weakness is the focus on only chicken-derived IgY. There has been significant work published with IgY from other avian species, e.g ducks and geese, which make similar high affinity IgY and lacking the same mammalian receptor interactions discussed above. The Introduction/Discussion should represent these IgY immunotherapeutics as well in the advancement of IgY.
The other comments require addressing but are minor in nature.
- In the Results 3.1 it is stated that that chick eggs yielded ~60 mg IgY/egg, yet in the Discussion stated 100 mg IgY/egg. Clarify which is correct
- In Figure 1, both the SDS-Page and WB clearly show a number of bands which apparently are IgY but do not match up with the heavy or light chain, or further dissociated piece of these chains. What are these? Please at least speculate since they are present in significant amounts.
- For Figure 2, the numbers of birds, etc. are unclear. What is the N utilized? It would be assumed that at any one time point, the titer, if generated from more than one bird would provide error bars? Please clarify this figure.
- Neutralization shown in Figure 5 is not clear until the highest concentration of IgY and yield the text says Igy can “potentially” neutralize at 12.5 ug/ml and above. Neutralization in Figure 5 appears to be more in alignment with Figure 6, discussed in #5 below, than as described in the text related to Figure 5. Either make this finding clear in the figure or remove the plaque assay pics. It also either is, or is not neutralizing at a concentration, please clarify this.
- The Results 3.6 describes Figure 6 as demonstrating a dose dependent neutralization by IgY. Figure 6 provides a sigmoid curve over the data, suggesting dose dependent findings. The curve is artificial at best with the data points actually being a plateau with a rapid elevation at the highest concentration. This, just as with Figure 5, supports the demonstrable neutralization only occurring at the highest tested concentrations.
- Figure 7 requires clarification on:
- Please identify the number of animals in each treatment. The histological analyses were preformed on 6 animals, but is n of 3/IgY type, or n of 6/IgY type. How many animals in for both MERS and control IgY each were included in the total study comprising Figure 7.
- Figure 7 Legend (A) suggests that the animals were treated with either MERS IgY or adjuvant. While this reviewer understands that it was likely control IgY from birds receiving only adjuvant, it should be clarified.
- Figure 7 C, to the naked eye the MERS IgY treated appears to have significantly more cellular infiltrate than does the control treated. The text says that the representative sections actually represent sections with the highest amount of cellular infiltrate. Therefore, is the appearance of Figure 7 C a misrepresentation in which the highest MERS IgY was higher than the highest control IgY treated, even though the mean infiltration was lesser in the MERS IgY? This needs to be clarified, and it would be much more informative for the reader to have truly representative sections from each treatment, not the highest cellular infiltrates observed.
- Throughout the paper, please ensure that Greek letters are appropriately presented.
Author Response
The authors present both very interesting and timely findings of the ability to rapidly raise MERS-CoV (MERS) S protein-specific IgY, present in large amounts. This MERS IgY recognized the S protein, and was neutralizing for the MERS-CoV virus both in vitro and in vivo. This is proposed to provide a passive immunotherapy both for MERS infections in humans and camels, as well as additional support for the potential of IgY immunotherapeutics.
Response: We thank the reviewer for the positive comment
- A major deficiency in the manuscript is the absence of discussion or hypothesis of the mechanism of protection by IgY. It has been published repeated that IgY is unique from mammalian immunoglobulins in that IgY does not bind to any mammalian FcRs, mammalian complement, rheumatoid factor, etc. While it can be argued that this provides an excellent niche for IgY because of the absence of inflammation associated with these various agents, this then makes the route of IgY neutralization unclear. This should be discussed in the Introduction (and/or possible the Discussion).
Response: We thank the reviewer for this comment. The following statement was added to the introduction describing the mode of action of IgY antibodies “IgY antibodies can neutralize viral infectivity by several mechanisms, either by blocking the attachment of the virus to host tissue; by preventing the membrane fusion or pro-moting the detachment of bound virus; by interfering with free virions; or by causing aggregation of virus particles resulting in virus immobilization (VanBlargan, Goo et al. 2016)”
- A second weakness is the focus on only chicken-derived IgY. There has been significant work published with IgY from other avian species, e.g ducks and geese, which make similar high affinity IgY and lacking the same mammalian receptor interactions discussed above. The Introduction/Discussion should represent these IgY immunotherapeutics as well in the advancement of IgY.
Response: We thank the reviewer for this comment, we have added the following paragraph in the introduction section “Despite the wide use of chicken for the production of IgY for research, immunization of other birds for IgY production has been also used with similar immunization protocols. Goose IgY antibodies were generated against dengue, West Nile virus, and Zika Virus and showed antiviral activity in mouse models for these infections (Fink, Williams et al. 2017). Other studies showed that goose and ducks IgY produced against Andes virus (which is the causative agent of the Hantavirus Pulmonary Syndrome) showed a prophylactic effect against the virus in infected mice and hamster (Brocato, Josleyn et al. 2012, Haese, Brocato et al. 2015). Ostriches immunized with the swine influenza virus were utilized to generate IgY antibodies against pandemic influenza virus IgY antibodies can be produced in large quantities (Tsukamoto, Hiroi et al. 2011). Quails were also used to generate anti-H. pylori IgY antibodies (Najdi, Brujeni et al. 2016). The wide use of chicken for the production of IgY antibodies might be due to the widespread and economic production of chicken eggs in large farms.”.
The other comments require addressing but are minor in nature.
- In the Results 3.1 it is stated that that chick eggs yielded ~60 mg IgY/egg, yet in the Discussion stated 100 mg IgY/egg. Clarify which is correct
Response: We apologize for this confusion, but the statement reported in the results describe the findings of this study while the statement in the discussion part was discussing the literature reported yield of IgY/egg. The statement in the discussion now reads “Previous studies showed that a single egg can yield up to 100 mg of total IgY”.
- In Figure 1, both the SDS-Page and WB clearly show a number of bands which apparently are IgY but do not match up with the heavy or light chain, or further dissociated piece of these chains. What are these? Please at least speculate since they are present in significant amounts.
Response: We thank the reviewer for this comment, we have added the following statement to the legend of figure 1 “Remaining bands might represent other antibodies or protein fragments of unknown origin.”
- For Figure 2, the numbers of birds, etc. are unclear. What is the N utilized? It would be assumed that at any one time point, the titer, if generated from more than one bird would provide error bars? Please clarify this figure.
Response: We apologize for such a confusion, but the number of hens used are stated in the methods section “2.1. Immunization of laying hens”. For the IgY titer, the antibodies were extracted from a pool of eggs of the same animal group at this specific time point, this sample pool was tested in triplicate and the OD values were very close that they did not produce error bars.
- Neutralization shown in Figure 5 is not clear until the highest concentration of IgY and yield the text says Igy can “potentially” neutralize at 12.5 ug/ml and above. Neutralization in Figure 5 appears to be more in alignment with Figure 6, discussed in #5 below, than as described in the text related to Figure 5. Either make this finding clear in the figure or remove the plaque assay pics. It also either is, or is not neutralizing at a concentration, please clarify this.
Response: In figure 5, the CPE shows protection of the cells up to a concentration of 12.5 µg/mL as evident by the lack of CPE up to this concentration. The figure was intended to show that there was no CPE up to the lowest concentration used compared to the virus control which shows very clear CPE. This is why we stated that the IC100 is <12.5 µg/mL; while in figure 6 the data are collected from real time RT-PCR testing which provide a more sensitive assay that can detect lower viral titers.
- The Results 3.6 describes Figure 6 as demonstrating a dose dependent neutralization by IgY. Figure 6 provides a sigmoid curve over the data, suggesting dose dependent findings. The curve is artificial at best with the data points actually being a plateau with a rapid elevation at the highest concentration. This, just as with Figure 5, supports the demonstrable neutralization only occurring at the highest tested concentrations.
Response: We thank the reviewer for this comment. Section 3.6 describes the results of the in vitro neutralization activity of the generated IgY. For more clarity, the section was revised to the following “The in vitro neutralization effect of the IgY Abs was examined by mixing different dilutions of the IgY Abs with MERS-CoV incubating for 1 hour at 37°C and then applying to the cells (as described in section 2.8). This approach showed a high neutralization effect on the virus at a 50% neutralizing concentration (NC50) of 51.42 mg/ml. The neutralization effect of the IgY Abs was assessed using real time RT-PCR of the cell cultures treated with different dilutions of the anti-MERS-CoV S IgY Abs, relative to the virus control cells (cells infected with the virus and untreated), which showed concentration-dependent inhibition of the virus (Figure 6). The log IgY concentration was plotted against the percentage of inhibition of each concentration and the NC50 was calculated following a nonlinear variable slope equation according to the equation: Y=100/(1+10^((LogIC50-X)*HillSlope)))”. Figure 6 was modified to show the actual line connecting the data points and the trend line, the legend was modified to “Figure 6. Virus neutralization titer of PCR-based virus neutralization test. IgY concentrations are represented in log10 (mg/ml). Red line connects the actual points while the black line represents the curve fitting line.”
- Figure 7 requires clarification on:
- Please identify the number of animals in each treatment. The histological analyses were preformed on 6 animals, but is n of 3/IgY type, or n of 6/IgY type. How many animals in for both MERS and control IgY each were included in the total study comprising Figure 7.
Response: We thank the reviewer for this comment. The details of testing animals are written in the methods section 2.9. The histological analysis was performed on 6 animals for each IgY.
- Figure 7 Legend (A) suggests that the animals were treated with either MERS IgY or adjuvant. While this reviewer understands that it was likely control IgY from birds receiving only adjuvant, it should be clarified.
Response: We thank the reviewer for this comment, all the legends of figure 7 components were changed to show “Adjuvant control” instead of “control IgY (Adjuvant)”
- Figure 7 C, to the naked eye the MERS IgY treated appears to have significantly more cellular infiltrate than does the control treated. The text says that the representative sections actually represent sections with the highest amount of cellular infiltrate. Therefore, is the appearance of Figure 7 C a misrepresentation in which the highest MERS IgY was higher than the highest control IgY treated, even though the mean infiltration was lesser in the MERS IgY? This needs to be clarified, and it would be much more informative for the reader to have truly representative sections from each treatment, not the highest cellular infiltrates observed.
Response: We apologize for the ambiguity. We reviewed the tissue samples and selected more appropriate lesion from an adjuvant control sample for Figure 7C. The previous figure 7C, left was the same site of viral antigen positive cells in the Figure 7E, left. We also modified the letters in the figures. Please see the attachment file.
- Throughout the paper, please ensure that Greek letters are appropriately presented.
Response: We apologize for this typing error; the Greek letters were revised throughout the manuscript.
Reviewer 2 Report
The MS is interesting; however, style, quality of presentation, and language must be improved. Below are my comments:
- It is not clear to me how these IgY antibodies would be more beneficial than humanized or human antibodies. Can these antibodies be used in humans without modification? This could lead to immune reactions in humans. The authors need to discuss this.
- The authors mentioned in the introduction and discussion that use of single antibody in treatment could result in escape viral mutants which is true; however, they did not comment on how useful would be IgY antibodies in that sense.
- Page 9 Figure 3: the authors indicated that the S protein was detected at 142kDa. However, it appears to me that the Mwt is 180kDa which is also consistent with the source of the protein being insect.
- I am not sure how figure 7A could be interpreted. Please clarify.
- When describing data in the results section, order of panels should be considered for example, start describing figure 7A then 7B then 7C etc..
- I am not sure what section 3.6 and figure 6 mean. This has to be rewritten.
- Images of figure 5 are not clear. You cannot tell any difference between images.
Author Response
The MS is interesting; however, style, quality of presentation, and language must be improved.
Response: The MS was submitted to Oxford English editing service for Language improvement.
Below are my comments:
- It is not clear to me how these IgY antibodies would be more beneficial than humanized or human antibodies. Can these antibodies be used in humans without modification? This could lead to immune reactions in humans. The authors need to discuss this.
Response: We thank the reviewer for this comment. The safety of IgY antibodies use are stated in the introduction section page 2 lines 77 to 80 as “IgYs are safer than antibodies from other species such as IgGs because they do not bind to human Fc receptors or fix mammalian complement components; hence, they do not trigger potentially dangerous immune responses” and more benefits of IgY use are stated in the discussion section page 14 lines 507 to 515 as “Chickens offer several advantages over conventional mammalian species in pro-ducing pathogen-specific antibodies. These include high rates of egg production, high IgY content per egg yolk (Rahman, Van Nguyen et al. 2013, Lee, Atif et al. 2017, So-masundaram, Choraria et al. 2020), and humane and non-invasive methods of collecting IgY Abs from eggs (Zhang, Chen et al. 2010, Spillner, Braren et al. 2012, Munhoz, Vargas et al. 2014). Clinical and laboratory data demonstrate that IgYs may offer a safe and effective tool for controlling and treating viral diseases. They may be used as a substitute for or essential complement to antimicrobials and vaccines (Karlsson, Kollberg et al. 2004, Nguyen, Tumpey et al. 2011, Amanat and Krammer 2020, Constantin, Neagu et al. 2020).”.
- The authors mentioned in the introduction and discussion that use of single antibody in treatment could result in escape viral mutants which is true; however, they did not comment on how useful would be IgY antibodies in that sense.
Response: We thank the reviewer for this comment, we have added to the discussion section the following statement “The use of IgY helps in reducing the risk of escape mutants as they target multiple epitopes making it harder for the virus to escape all the targeted positions”.
- Page 9 Figure 3: the authors indicated that the S protein was detected at 142kDa. However, it appears to me that the Mwt is 180kDa which is also consistent with the source of the protein being insect.
Response: We thank the reviewer for this comment. We used the MERS-CoV S protein from Sino biologicals (Cat: 40069-V08B). The distance between the bands in the SDS page figure is not far enough to discriminate between the bands of the ladder at 135 and 180kDa.
- I am not sure how figure 7A could be interpreted. Please clarify.
Response: We apologize for the ambiguity. Legend of figure was modified as follows: Viral titers in lung homogenates of MERS-CoV infected mice at 1, 3, 5 days after inoculation (n = 4 per group). Mice were treated with anti-S IgY antibodies or adjuvant only. The detection limit was 10^1.5 TCID50/g of tissue. There is no statistical significance by Bonferroni’s multiple comparisons test.
- When describing data in the results section, order of panels should be considered for example, start describing figure 7A then 7B then 7C etc..
Response: We thank the reviewer for this comment. The order of statements has been arranged according to the order of figure 7 sections.
- I am not sure what section 3.6 and figure 6 mean. This has to be rewritten.
Response: We thank the reviewer for this comment. Section 3.6 describes the results of the in vitro neutralization activity of the generated IgY. For more clarity, the section was revised to the following “The in vitro neutralization effect of the IgY Abs was examined by mixing different dilutions of the IgY Abs with MERS-CoV incubating for 1 hour at 37°C and then applying to the cells (as described in section 2.8). This approach showed a high neutralization effect on the virus at a 50% neutralizing concentration (NC50) of 51.42 g/ml. The neutralization effect of the IgY Abs was assessed using real time RT-PCR of the cell cultures treated with different dilutions of the anti-MERS-CoV S IgY Abs, relative to the virus control cells (cells infected with the virus and untreated), which showed concentration-dependent inhibition of the virus (Figure 6). The log IgY concentration was plotted against the percentage of inhibition of each concentration and the NC50 was calculated following a nonlinear variable slope equation according to the equation: Y=100/(1+10^((LogIC50-X)*HillSlope)))”. Figure 6 was modified to show the actual line connecting the data points and the trend line, the legend was modified to “Figure 6. Virus neutralization titer of PCR-based virus neutralization test. IgY concentrations are represented in log10 (mg/ml). Red line connects the actual points while the black line represents the curve fitting line.”
- Images of figure 5 are not clear. You cannot tell any difference between images.
Response: In figure 5, there was no CPE which shows protection of the cells up to a concentration of 12.5 µg/mL as evident by the lack of CPE up to this concentration. The figure was intended to show that there was no CPE up to the lowest concentration of IgY antibodies used compared to the virus control which shows very clear CPE. This is why we stated that the IC100 is <12.5 µg/mL.
Round 2
Reviewer 2 Report
All comments have been addressed.